# Preferential Ribosome Loading on the Stress-Upregulated mRNA Pool Shapes the Selective Translation under Stress Conditions

**DOI:** 10.3390/plants10020304

**Published:** 2021-02-05

**Authors:** Yan Chen, Min Liu, Zhicheng Dong

**Affiliations:** Guangzhou Key Laboratory of Crop Gene Editing, Innovative Center of Molecular Genetics and Evolution, School of Life Sciences, Guangzhou University, Guangzhou Higher Education Mega Center, 230 Waihuanxi Road, Guangzhou 510006, China; Yanchen@gzhu.edu.cn (Y.C.); minl@gzhu.edu.cn (M.L.)

**Keywords:** *Arabidopsis thaliana*, Ribo-seq, transcriptome, translatome, preferential translation, stress response

## Abstract

The reprogramming of gene expression is one of the key responses to environmental stimuli, whereas changes in mRNA do not necessarily bring forth corresponding changes of the protein, which seems partially due to the stress-induced selective translation. To address this issue, we systematically compared the transcriptome and translatome using self-produced and publicly available datasets to decipher how and to what extent the coordination and discordance between transcription and translation came to be in response to wounding (self-produced), dark to light transition, heat, hypoxia, Pi starvation and the pathogen-associated molecular pattern (elf18) in *Arabidopsis.* We found that changes in total mRNAs (transcriptome) and ribosome-protected fragments (translatome) are highly correlated upon dark to light transition or heat stress. However, this close correlation was generally lost under other four stresses analyzed in this study, especially during immune response, which suggests that transcription and translation are differentially coordinated under distinct stress conditions. Moreover, Gene Ontology (GO) enrichment analysis showed that typical stress responsive genes were upregulated at both transcriptional and translational levels, while non-stress-specific responsive genes were changed solely at either level or downregulated at both levels. Taking wounding responsive genes for example, typical stress responsive genes are generally involved in functional categories related to dealing with the deleterious effects caused by the imposed wounding stress, such as response to wounding, response to water deprivation and response to jasmonic acid, whereas non-stress-specific responsive genes are often enriched in functional categories like S-glycoside biosynthetic process, photosynthesis and DNA-templated transcription. Collectively, our results revealed the differential as well as targeted coordination between transcriptome and translatome in response to diverse stresses, thus suggesting a potential model wherein preferential ribosome loading onto the stress-upregulated mRNA pool could be a pacing factor for selective translation.

## 1. Introduction

Plant growth and development can be profoundly affected by various stress conditions. To elucidate the stress response is thus of great importance, especially for sessile plants. Organisms have evolved sophisticated mechanisms comprising multiple biological processes to combat stresses, such as selective gene expression [1]. Specifically, the amount and activity of a certain protein are balanced outputs of mRNA transcription, processing, modification, and decay as well as protein translation, location, modification and degradation [2,3,4]. Until recently, the mRNA level has been used to predict corresponding protein abundance, based on the assumption that mRNA changes must be accompanied by coordinated changes in the protein. However, emerging evidence from yeasts, plants, and mammals has shown that translation efficiency, rather than mRNA abundance, contributes most to the final protein amount [3,5,6,7] (Appendix A). Although technologies for the mass spectrometric analysis of proteins have been intensively developing, the resolution and sensitivity is still limited compared to nucleic acid sequencing methods. More importantly, being the final outputs of gene expression, the protein levels can neither reflect the real-time cellular response nor provide information regarding molecular mechanisms underlying the translation regulation.

Given the central role of translation, it is critically important to monitor the process of protein synthesis in a direct and delicate way. Among the methodologies for interrogating translatome, polysome-associated mRNA-seq, namely polysome profiling, can measure the abundance of mRNA molecules present in polysomes [8], while critical translational details, such as the ribosome position and density depicting the ribosome dynamics, are generated by ribosome profiling, a revolutionary tool widely applied in various organisms to map the ribosome-protected fragments (RPFs) [5,9,10,11,12,13,14]. Explicitly, the number of RPFs per gene reflects the extent of ribosome loading, which then shapes the translatome. Therefore, the term translatome is often referred to RPFs. Additionally, in combination with ribosome profiling and conventional RNA-sequencing data from the same sample, it enables the evaluation of the translational efficiency for mRNAs from individual genes on a genomewide level, defined as the ratio of RPF to mRNA abundance as to each protein-coding sequence.

Recent years have also seen a substantial interest in dissecting the relationship between transcriptional and translational regulations in yeasts, plants and mammals under diverse stresses or across different tissues [15,16,17,18,19,20,21,22,23,24]. However, most of these studies assess the mRNA translation through polysome profiling, which cannot measure the exact number of ribosomes loading on polysome-associated mRNAs for each gene [19,20,21,22,23]. By contrast, a few studies comprehensively investigated the translatome based on ribosome footprints [15,16,17,18,24]. For instance, one study in yeast observed that under severe stress conditions such as amino acid depletion or osmotic shock, changes in the transcriptome (mRNAs) correlated well with those in translatome (RPFs), while hardly any or relatively poor correlations were found between these two levels in response to mild stresses induced by calcofluor-white and menadione [15]. Another study in yeast revealed the homodirectional regulation of transcriptome (mRNAs) and translatome (RPFs) in response to hydrogen peroxide [24]. Moreover, the comparative transcriptome (mRNAs) and translatome (RPFs) analysis in human primary hepatocytes under hypoxia demonstrated that translational responses preceded and were predominant over transcriptional responses [16]. Remarkably, a recent study [18], using Ribo-seq and matched RNA-seq data to compare transcriptome (mRNAs) and translatome (RPFs) in three primary organs across five representative mammals and a bird, uncovered the differential regulation and coevolution between the two expression layers, thus providing a substantial insight into their interplay in mammalian organs. These studies revealed the complexity of coordination between transcriptome (mRNAs) and translatome (RPFs) both in yeasts and mammals. In plants, plenty of stress-responsive transcriptomes (mRNAs) and translatomes (RPFs) have been profiled [12,25,26,27,28]; however, in contrast to the studies in yeasts and mammals, relatively few comparative analyses were conducted to examine the interrelation between them. 

Here in this study, we systematically investigated the transcriptome and translatome in response to wounding as well as other five stresses in *Arabidopsis*, including dark to light transition [12], heat [25], hypoxia [26], Pi starvation [27] and the pathogen-associated molecular pattern (elf18) [28]. This led to the finding that transcription and translation belong to different layers of regulation with intrinsic connection. Changes in transcriptome and translatome are differentially correlated upon distinct stress conditions, suggesting that ribosome loading is another fundamental level of gene expression regulation after the transcriptional response. Moreover, genes specifically responding to the imposed stress are often upregulated at both transcriptional and translational levels. Overall, our results suggest that stress-induced selective translation is mainly shaped at translational level through preferential ribosome loading on the stress-upregulated mRNAs.

## 2. Results

### 2.1. Stress-Dependent Correlation between Changes in Transcriptome and Translatome

Plant responses to wounding stress have been extensively studied at transcriptional level. However, we lack a genome-wide understanding of the translational response and how it coordinated with the transcriptome. To address this issue, we measured the abundance of total and polysome associated mRNAs as well as mapping the number and position of translating-ribosomes in the control (C) and wounded (W) *Arabidopsis* leaf tissue (Appendix A). All experiments were performed on three biological replicates with high reproducibility (Appendix A). As shown by principal component analysis (PCA, Appendix A), the expression profiles were highly reproducible between biological replicates, with evident patterns before and after wounding and across total mRNAs, polysomal mRNAs and RPFs. The majority of the RPFs were assigned to CDS regions (C 67.4%, W 73.2%), with far smaller proportions to 5′ UTR (C 3.5%, W 3.7%) and 3′ UTR (C 3.0%, W 3.0%), which confirmed the strong association of ribosomes with the CDS (Appendix A). Given the fact that translating ribosomes advance three nucleotides during each decoding cycle, the read density of RPFs 5′-end at each nucleotide around the annotated start or stop codon (100-nt upstream and downstream) displayed strong 3-nt periodicity and moreover, consistent with previous studies in response to dark to light transition and Pi starvation [12,27], two observable peaks were found upstream of the start and stop codons under both control and stress conditions, illustrating the occurrence of ribosome stalling during translation initiation or termination (Appendix A). Nonuniform ribosome distribution along an mRNA was also observed in other translatome studies interpreting immune (elf18), heat, or hypoxia response; however, the ribosome movement was not strongly periodic and higher ribosome density was found at the start codon but not upstream of the start or stop codon, probably due to the imposed stress or the adopted experimental strategy (Appendix A) [25,26,28].

To ask how wounding affects the gene expression at both transcriptional and translational levels simultaneously, correlation analysis was conducted between the transcriptome and translatome. In contrast to the strong correlation among the overall levels of total mRNAs (mRNA), polysome-associated mRNAs (Poly) and ribosome-protected fragments (RPF) (Figure 1A), the fold changes (wounding/control) in total mRNAs and RPFs were not so well correlated (*r* = 0.69; Figure 1B); however, relatively high correlation was still found between the fold changes in total mRNAs and polysome-associated mRNAs (Poly) (*r* = 0.82; Figure 1C), indicating that upon wounding, transcribed mRNAs were proportionally engaged with the translation machinery, while ribosomes would not evenly load onto the polysome-associated mRNAs by the same proportion. A similar analysis was performed for the publicly available data produced from *Arabidopsis* treated with elf18, heat, hypoxia, dark to light transition and Pi starvation (Table 1) [12,25,26,27,28]. Differential correlations between fold changes in total mRNAs and RPFs were unveiled upon distinct stresses (Figure 1D,E, Appendix A), where the high correlation between total mRNAs and RPFs changes upon dark to light transition (*r* = 0.91; Figure 1D) was largely lost during elf18-induced immune response (*r* = 0.23; Figure 1E).

Moreover, different stresses selectively altered the translational efficiency (TE) of a subset of mRNAs (Appendix A; TE, ratio of RPF abundance to mRNA abundance). Under wounding, 581 genes had significantly changed TE, of which 397 (68.5%) were downregulated. The general decline or no change in TE is due to the fact that the majority of mRNAs are not loaded with ribosomes in proportion to the level they increase upon stress. As shown, wound-induced fold changes in mRNA negatively correlated with those in TE (*r* = −0.55) (Figure 1F), which was true for elf18 (*r* = −0.56), hypoxia (*r* = −0.17), heat (*r* = −0.16) and Pi starvation (*r* = −0.27) (Appendix A). Under dark to light transition, little correlation was observed between the fold changes in mRNA and TE (*r* = 0.09) (Appendix A). Thus, both transcription and ribosome loading are involved in the regulation of protein synthesis, and the mRNA level should not be taken as the sole determinant for tuning protein production. Altogether, these results revealed distinct levels of correlation between changes in transcriptome and translatome under different stresses and moreover, stress-specific ribosome loading is a critical layer for the gene expression regulation. 

### 2.2. Divergent and Coordinated Changes in Transcriptome and Translatome upon Wounding

Then we asked which genes were significantly regulated upon wounding on their total mRNAs, polysome-associated mRNAs and ribosome-protected fragments (RPFs) levels (Appendix A). Figure 2A shows that most genes altered in polysome association possess according changes in their mRNA levels, while a relatively smaller proportion of genes with changed ribosome loading (RPFs) have the same trend of mRNA variation. We further performed the correlation analysis between total mRNAs, polysome occupancy and ribosome loading (RPFs) data sets, particularly for genes that are up- or downregulated at either of the three levels, and indeed observed a much closer correlation between total mRNAs and polysome occupancy than between total mRNAs and ribosome loading (RPFs) or between polysome occupancy and ribosome loading (RPFs) (Figure 2B). Intriguingly, for genes up- or downregulated at all three levels, the correlation was relatively high and comparable between any two of the three levels, especially for the upregulated genes (Figure 2C). These results revealed both divergent and coordinated changes of transcriptome (mRNAs) and translatome (RPFs) under wounding, which signified the preferential reprogramming of translatome through adjusting the ribosome loading on the regulated mRNAs. Specifically, genes showing both transcriptome (mRNAs) and translatome (RPFs) changes revealed the coordinated reactions at these two steps upon wounding, exemplified by ERD9 (Early-Responsive to Dehydration 9) and FBA5 (Fructose-Bisphosphate Aldolase 5) with proportional changes in transcription and translation (Figure 2D). DALL1 (DAD1-like Lipase 1) is another representative showing increased mRNA level and polysome association, but unchanged ribosome loading under wounding. HSC70-1 (Heat Shock Cognate Protein 70-1), encoding a member of the heat shock protein 70 family, illustrates wounding-induced upregulation of mRNA level and ribosome loading without a marked variation in polysome association, which unveiled the regulation mainly through adjustment of ribosome loading on the mRNA. TCTP1 (Translationally Controlled Tumor Protein 1), homologous to translationally controlled tumor protein (TCTP) from *Drosophila* involved in the TOR signaling pathway, reflects the situation where ribosome loading increased upon wounding in the absence of a significant change in mRNA level and polysome association. These varying patterns of gene-specific translational control suggest that preferential ribosome loading onto a given mRNA should be a gateway in tuning protein synthesis in response to wounding.

These results indicate that a subset of transcriptionally regulated mRNAs was preferentially regulated at translational level upon wounding, which is consistent with the observation from yeast cells [29]. However, whether transcriptionally regulated but translationally unchanged mRNAs aggregate into stress granules requires further validation [30]. Meanwhile, different patterns of coordination between transcription and translation take place among different genes under wounding. Therefore, gene expressions display both divergent and coordinated changes at transcriptional and translational levels during stress response. More importantly, the reprogramming of gene expression is mainly orchestrated at translational level by modulating ribosome loading but not polysome association, which means that to what extent the mRNAs are translated may ultimately depend more on how many ribosomes load on the mRNAs rather than how many mRNAs are occupied by ribosomes, as evidenced by our results (Figure 2B). In theory, as long as the necessary distance is maintained, the more ribosomes load on the mRNAs, the more actively the mRNAs are translated.

### 2.3. Targeted Translational Control under Different Stresses

To answer whether genes coordinately regulated at both transcriptional and translational levels are especially important for specific stress response, we compared the translational efficiency (TE) of genes altered merely at transcriptional (mRNAs) or translational (RPFs) level as well as those that changed at both levels in response to different stresses. It is interesting to find that TE volatility for genes regulated at both levels is locked in a narrow range, compared with that for genes altered solely at either level, suggesting that more stringent translational control is executed on this group of genes (Figure 3A). It is worth noting that when analyzed for enriched Gene Ontology terms, genes specifically dealing with the imposed stress were often coordinately regulated, or, more precisely, upregulated at both levels; however, those changed merely at mRNA or RPF level or downregulated at both levels are not typical genes to dispose of the corresponding stress (Figure 3B,C, Appendix A). Taking the wounding responsive genes as an example, the overrepresented functional terms for these gene sets showed distinct patterns with little overlapping, and genes upregulated at both levels were generally involved in typical biological processes for wounding responses like response to wounding, response to water deprivation and response to jasmonic acid (Figure 3B,C). This makes explicit the concept that the increment of both mRNA and RPF levels for a specific set of genes seems to be a common mechanism to ensure a more targeted response against each stress. In addition, this analysis exposed multifaceted gene regulation by invoking genes related to different stress responses other than wounding, including immune response, programmed cell death, transport, response to chitin, ethylene, heat, oxidative stress, ABA, etc. (Figure 3B,C, Appendix A). 

Subsequently, we sought to further explore whether there are genes coordinately regulated at both transcriptional and translational levels across different stresses, which would be the core responsive genes that are crucial for stress response and plant survival. Although there are no genes in common responsive to all six stresses, a subset of genes was identified to be induced at both levels by multiple stresses (Appendix A). Among these, the expression of WRKY family genes, encoding key transcription factors for stress response and development, fluctuated upon exposure to wounding, heat, hypoxia or elf18. In addition, heat shock proteins were also involved in diverse stress responses, especially HSP90, which is upregulated by five distinct stresses including wounding, heat, hypoxia, dark to light transition and Pi starvation, suggesting its more general and essential function during stress response.

## 3. Discussion

In this study, we systematically investigated the correlation between changes in transcriptome and translatome upon wounding and other stress conditions [12,25,26,27,28], revealing the complexity of coordination between these two levels as well as the selective translation resulting from preferential ribosome loading on the upregulated mRNAs. Moreover, we provided a catalog of core responsive genes, which code for proteins that may act as key driving forces of the multiple stress response. All these findings highlight a potential mechanism for how during stress response, targeted gene expression takes place at both transcriptional and translational levels but mainly shapes up at translational level through the preferential ribosome loading onto the upregulated mRNAs (Figure 4). However, the determinants specifying the stress-dependent ribosome loading and preferential translation remains to be explored.

Herein, we have established an approach to rapidly assess the levels of total mRNAs, polysome-associated mRNAs and ribosome-protected fragments, whereby the transcriptional and translational profiles in *Arabidopsis thaliana* under wounding were obtained. Specifically, we used size exclusion rather than conventional density gradient based methodology to fractionate polysomes from the crude polysome extract. In fact, the basic principle for these two methods is essentially the same, both separating polysomes from monosomes according to the molecular weight. Previous studies demonstrated that the molecular weight of eukaryotic ribosomes ranges from 3.3 MD to 4.3 MD [31,32] and most of the ORFs are loaded with more than three ribosomes on average during active translation [8,33,34]. Based on this information, Syphacryl S-400 HR with an upper exclusion limit of 8 MD was selected to purify the polysomes. Thus, polysomes with a molecular weight equivalent to at least three ribosomes (9.9 MD to 12.9 MD) are eluted just after the void volume, accounting for approximately 30% of the total column volume (packed bed). The packed column volume can be adjusted according to the loading volume, which is generally 15–20% of the total column volume. After column equilibration and sample loading, the polysomes were collected by gravity flow. Our alternative approach is independent of specific equipment, requires fewer starting materials and shorter operative time and most importantly, yields robust results with good resolution.

Comparative analysis in this study led to the finding that transcriptome (mRNAs) and translatome (RPFs) were differentially coordinated under distinct stresses and eukaryotic gene expression was controlled by multilayered regulations, including not only the transcription but also the more energy-intensive translation. Specifically, we found that changes in transcriptome (mRNAs) and translatome (RPFs) were highly correlated upon dark to light transition (*r* = 0.92) or heat (*r* = 0.79) treatments, while moderate or even poor correlations were observed under wounding (*r* = 0.69), hypoxia (*r* = 0.63), Pi starvation (*r* = 0.63) or elf18 (*r* = 0.23) induced stresses, which raises a question on how to explain the observed differences among distinct stresses. In this regard, studies interpreting yeast response to high salinity and human primary hepatocytes response to hypoxia should shed some light [16,23]. It has long been believed that unfavorable conditions would induce global translational repression as well as selective translation [35]. A time course analysis through polysome profiling of yeast response to 1 M NaCl uncovered that global translational inhibition peaks at 1 h after the onset of high salinity and recovers within 5 h, while the transcriptional levels of salt-related genes only start increasing at 1 h [23]. This indicates that the most drastic translational change occurred at 1 h, at which time point the transcriptional levels only changed to a much lesser extent, as described by earlier studies indicating that high salinity would elicit a delayed transcriptional response [36,37]. Despite the low resolution of polysome profiling, this parallel study of mRNA transcription and translation suggested the possibility for discordant responses with respect to transcription and translation under high salinity. Consistently, the study in human primary hepatocytes revealed that translational responses appeared earlier than transcriptional responses and moreover, a closer correlation between transcriptome (mRNAs) and translatome (RPFs) upon 240 min hypoxia exposure relative to 30 min hypoxia exposure. Taken together, the above results indicate that gene expression changes at both transcriptional and translational levels under stresses, while whether or not the two responses are well coordinated depends at least in part on the sampling time after the stress imposition. Plant response to wounding occurs over a time scale ranging from several minutes to a few hours [38]. In this study, we selected the time point of 3 h after wounding to avoid the very likely delayed transcriptional changes for some genes during early response within minutes. Nevertheless, to further elucidate the mechanism underlying varied correlations between transcriptome (mRNAs) and translatome (RPFs) upon distinct stresses in *Arabidopsis* needs comprehensive time course analysis. 

In contrast to the translational repression that allows for energy conservation and cell recovery, selective translation of upregulated mRNAs through preferential ribosome loading are critically important for corresponding stress response, as evidenced by our results. This phenomenon is referred to as potentiation (coordinately regulated transcription and translation), which is believed to allow amplification of transcriptional changes at translational levels to generate a more robust and faster response [21]. Specially, our study further demonstrated that these coupregulated genes are specifically standing up against the imposed stress. Furthermore, it is worth noting that our study also adds to the evidence that ribosome profiling outperforms polysome association at assessing the translational activity of individual genes. This occurs because polysome association only measures the number of mRNA molecules occupied by polysomes and ribosome profiling deciphers the exact number of ribosomes loading on polysome-associated mRNAs for each gene, which then reflects the translational activity.

Moreover, we identified a number of core responsive genes that are coregulated at both transcriptional and translational levels by multiple stresses. Particularly, we highlighted the presence of two types of core responsive genes that were regulated at both levels by at least four stresses, the WRKY family of genes and genes encoding heat shock proteins (HSPs). WRKY transcription factor genes, known as the largest gene family in higher plants, have been reported in many species including *Arabidopsis*, rice, tomato, cotton, pineapple and wild strawberry [39]. Massive studies have indicated the regulatory role of WRKY gene members that confers biotic as well as abiotic stress tolerance to crop plants [40]. Heat shock proteins (HSPs), often performing chaperone functions [41], express not only in response to various stresses [42,43,44,45], but also exhibit developmental and clinical significances [46,47,48]. Given the broad spectrum of stress tolerance, these two types of hub genes have far-reaching implications for crop improvement and molecular breeding.

## 4. Materials and Methods

### 4.1. Plant Growth and Treatment

*Arabidopsis thaliana* (Col-0) were grown on soil at 22 °C under a 16-h light/8-h dark cycle with ~50% humidity. Wounding treatment was introduced by crushing 21-d rosette leaves for several times with forceps, which effectively wounded ~30% of the leaf area. Note that for each sample seven to eight 21-d rosette leaves per plant and a total of four plants underwent wounding treatment. Subsequently, control (unwounded) and wounded plants were incubated under the same light and humidity conditions for 3 h. Leaves were harvested, immediately frozen in liquid nitrogen, and stored at −80 °C.

### 4.2. Preparation of Total, Polysome-Associated mRNA and Ribosome-Protected Fragments

We used 150 μL of pulverized frozen leaf tissue for polysome isolation by adding 1 mL freshly prepared polysome extraction buffer [200 mM Tris-HCl (pH 9.0), 200 mM KCl, 36 mM MgCl_2_, 25 mM EGTA, 50 μg/mL cycloheximide, 50 μg/mL chloramphenicol, 1% Triton X-100 (vol/vol), 1% Tween-40 (vol/vol), 1% Nonidet P-40 (vol/vol), 1% polyoxyethylene lauryl ether (vol/vol), 1% polyoxyethylene 10 tridecylether (vol/vol), 5 mM DTT, 1 mM phenylmethanesulfonylfluoride and thawing on ice for 10 min with occasional mixing [26]. Cell debris was removed by centrifugation at 4 °C, 17,000× *g* for 10 min. We saved 10% volume (100 μL) of the supernatant for total RNA extraction using TRIzol (Invitrogen, Carlsbad, USA). The remaining ~800 μL supernatant was divided into two 400 μL copies and each was fractionated by size exclusion on 2 mL self-packed columns of Sephacryl S-400 High Resolution (17-0609-10, GE Healthcare, Marlborough, USA) to produce a fraction consisting predominantly of polysomes, which is eluted just after the void volume. For a well packed column, the void volume accounts for approximately 30% of the total column volume [49]. The packed columns were fully equilibrated by RNase I digestion buffer [20 mM Tris-HCl (pH 8.0), 140 mM KCl, 35 mM MgCl_2_, 50 μg/mL cycloheximide] immediately before loading the supernatant. One copy of the obtained polysomes was directly used to isolate polysome-associated RNA by TRIzol, and the other was applied to nuclease digestion to obtain ribosome-protected fragments (RPFs) by adding 0.8 U/μL RNase I (Ambion, Carlsbad, USA) and 2 U/μL DNase I (Turbo DNase Ambion, Carlsbad, USA). Digestion was conducted for 45 min at room temperature with gentle rotation. RPFs were purified using TRIzol.

### 4.3. Library Construction and Sequencing

poly(A)^+^ mRNA was selected by Oligo (dT)_20_ Dynabeads (Invitrogen) from total and polysome associated RNA according to the manufacturer’s instructions for two rounds and applied to mRNA-seq libraries preparation using NEBNext Ultra^TM^ II Directional RNA Library Prep Kit for Illumina (NEB, Ipswich, USA). The RPFs were further separated on a 15% TBE-Urea denaturing polyacrylamide gel and recovered by size-excision (20–40 nt). After rRNA depletion (MRZPL1224, Illumina, San Diego, CA, USA) and PNK treatment (M0201L, NEB), RPFs were processed into Ribo-seq libraries using the NEXTflex^TM^ Small RNA-Seq Kit v3 (Bioo Scientific, Austin, TX, USA). Equal molar ratios of the barcoded libraries were pooled for Illumina pair-end 150 bp sequencing.

### 4.4. Bioinformatic Analysis

All data sets were subjected to quality assessment using the software FASTQC (https://www.bioinformatics.babraham.ac.uk/projects/fastqc/ (accessed on 15 October 2020)). Raw reads were trimmed by Cutadapt to remove adaptors [50], and too short reads were filtered before alignment to TAIR10 genome using the split-aware aligner STAR [51]. Uniquely mapped reads were then assigned to genomic features (5′ UTR, CDS, 3′ UTR, exons) defined by the latest Araport11 genome annotation using featureCounts [52]. Read counts were used to calculate Reads Per Kilobase of transcript per Million mapped reads (RPKM) and Transcripts Per Million (TPM), respectively. Log_2_TPM were used for calculating the Pearson correlation coefficient among samples in Appendix A and PCA analysis in Appendix A. The average TPM for three biological replicates were calculated and the log_2_(average TPM) was used for the correlation analysis in Figure 1A. Read counts were used as the input for calculating changes in transcriptome, translatome and translational efficiency (TE) by DESeq2 [53]. Significant changes in total mRNA, polysomal mRNA and ribosome-protected fragments (RPFs) were defined by RPKM ≥ 1 in at least one sample, |log_2_FC| ≥ 1, and adjusted *p*-value (padj) ≤ 0.05. Log_2_FC of genes with RPKM ≥ 1 in at least one sample were used to perform the correlation analysis in Figure 1B–F and Appendix A. TE changes can be calculated by ratios of ratios for two conditions following the formula [(Ribo_stress/mRNA_stress)/(Ribo_control/mRNA_control)] according to systemPipeR Workflow for Ribo-seq [54]. Significant TE changes were defined by RPKM ≥ 1 in at least one sample and |z| ≥ 1.5. z represents the z-score of log_2_(TE_FC_) calculated from the formula [z = (x − μ)/σ], where x is the raw value of each log_2_(TE_FC_), μ is the mean of all log_2_(TE_FC_), and σ is the standard deviation of all log_2_(TE_FC_). Read counts were used as the input for metagene analysis. Subsequent analysis such as Gene Ontology (GO) enrichment and presentation of statistical data were conducted and plotted through R (Appendix A). 

## Figures and Tables

**Figure 1 plants-10-00304-f001:**
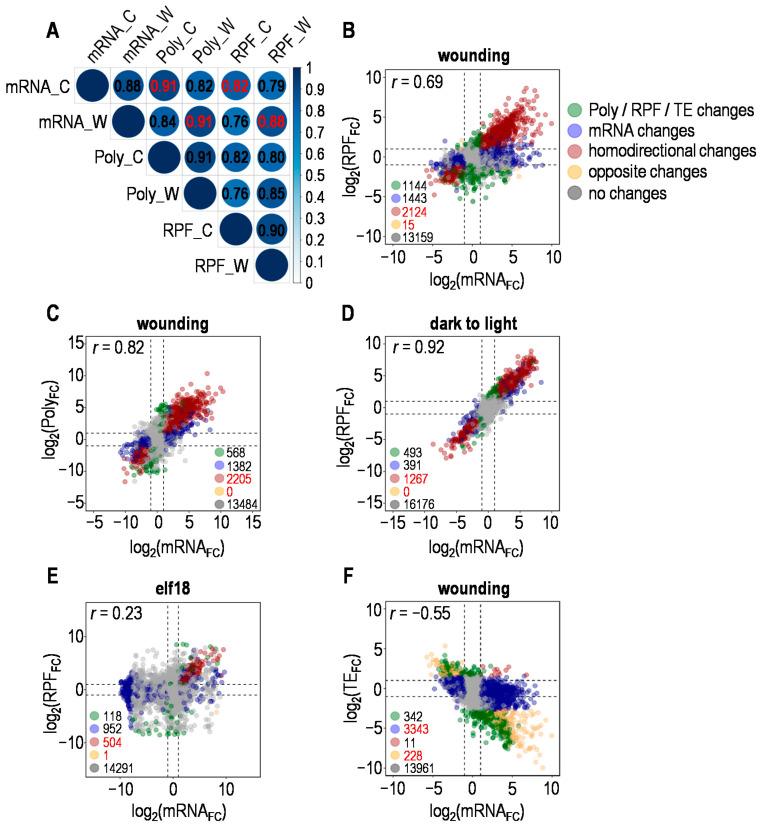
Correlations of transcriptome and translatome under indicated stresses. (**A**) Correlation of total mRNAs (mRNA), polysome-associated mRNAs (Poly) and ribosome-protected fragments (RPF) levels before and after wounding. The base-2 logarithm (log_2_) of the Transcripts Per Million (TPM) was used to calculate the Pearson correlation coefficient between samples; color-coded number is the Pearson correlation coefficient between two samples; C for control, W for wounding. (**B**) Relationship between fold changes in total mRNAs (mRNA_FC_) and ribosome-protected fragments (RPF_FC_) introduced by wounding (DESeq2; *n* = 3 for total mRNA-seq and/or Ribo-seq). Color-coded dots represent genes with available values of mRNA_FC_ and RPF_FC_ under wounding; significant total mRNA or RPF changes were defined as described in Materials and Methods; dotted lines indicate the values of corresponding log_2_FC equal to −1 and 1; Pearson correlation coefficient *r* was shown. (**C**) Relationship between fold changes in total mRNA (mRNA_FC_) and polysome-associated mRNA (Poly_FC_) introduced by wounding (DESeq2; *n* = 3 for total mRNA-seq and/or polysome-associated mRNA-seq). Color-coded dots represent genes with available values of mRNA_FC_ and Poly_FC_ under wounding; other details were as described for (**B**). (**D**) Relationship between fold changes in total mRNAs (mRNA_FC_) and ribosome-protected fragments (RPF_FC_) introduced by dark to light transition. Details were as described for (**B**). (**E**) Relationship between fold changes in total mRNAs (mRNA_FC_) and ribosome-protected fragments (RPF_FC_) introduced by pathogen-associated molecular pattern (elf18). Details were as described for (**B**). (**F**) Relationship between fold changes in total mRNAs (mRNA_FC_) and translational efficiency (TE_FC_) introduced by wounding (DESeq2; *n* = 3 for mRNA-seq and/or Ribo-seq; TE, ratio of RPF abundance to mRNA abundance). Color-coded dots represent the same set of genes as in (**B**); significant TE changes were defined as described in Materials and Methods.

**Figure 2 plants-10-00304-f002:**
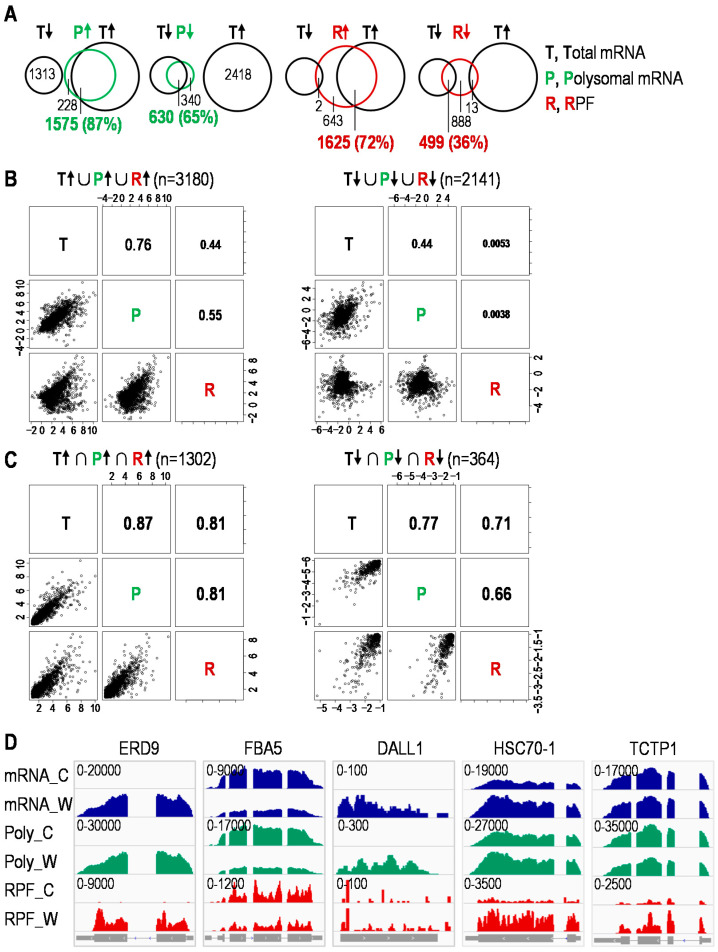
Wound-induced coordination and divergence of transcriptome and translatome. (**A**) Venn diagrams depicting the number of genes with up or downregulated total mRNA, polysome-associated (polysomal) mRNA or ribosome-protected fragments (RPF) in response to wounding. Green numbers represent genes with homodirectional changes in total mRNA (T) and polysomal mRNA (P); red numbers represent genes with homodirectional changes in total mRNA (T) and ribosome-protected fragments (R); arrows indicate up or down changes. (**B**) Relationship between total mRNA (T), polysome occupancy (P) and ribosome loading (R) data sets for genes up- or downregulated at either of the three levels (DESeq2; *n* = 3 for total mRNA-seq, polysome-associated mRNA-seq and/or Ribo-seq). Significant changes were defined as described in Materials and Methods; Pearson correlation coefficient *r* was shown. (**C**) Relationship between total mRNA (T), polysome occupancy (P) and ribosome loading (R) data sets for genes up- or downregulated at all three levels. (**D**) Normalized read coverage of total mRNA (mRNA), polysomal mRNA (Poly) and RPF for selected genes. C for control, W for wounding; gene structure was shown at the bottom.

**Figure 3 plants-10-00304-f003:**
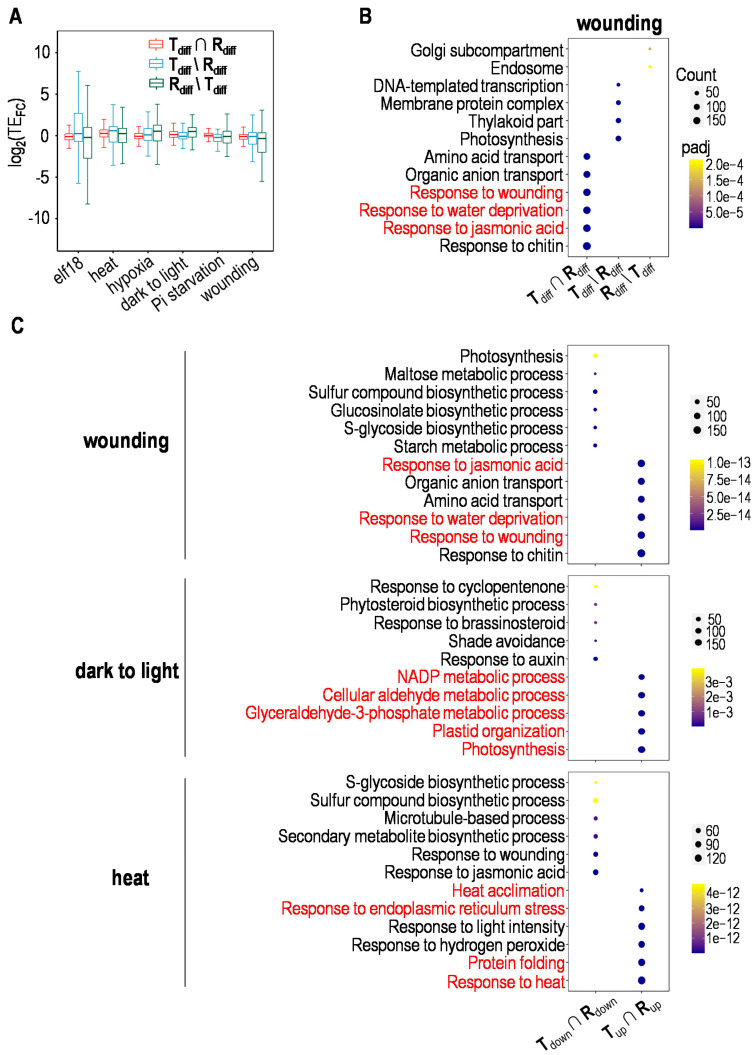
Targeted translational response to wounding. (**A**) Translational efficiency (TE) of genes altered merely at transcriptional or translational level as well as those changed at both levels in response to different stresses. T_diff_ ∩ R_diff_ represents genes differentially expressed at both transcriptional (mRNA) and translational (RPF) levels; T_diff_\R_diff_ represents genes differentially expressed solely at transcriptional level; R_diff_\T_diff_ represents genes differentially expressed solely at translational level. (**B**) Functional categories of differentially expressed genes under wounding. Details were as described for (**A**). (**C**) Functional categories of up- or downregulated genes under wounding, dark to light transition or heat. T_up_ ∩ R_up_ represents genes upregulated at both transcriptional (mRNA) and translational (RPF) levels; T_down_ ∩ R_down_ represents genes downregulated at both transcriptional and translational levels.

**Figure 4 plants-10-00304-f004:**
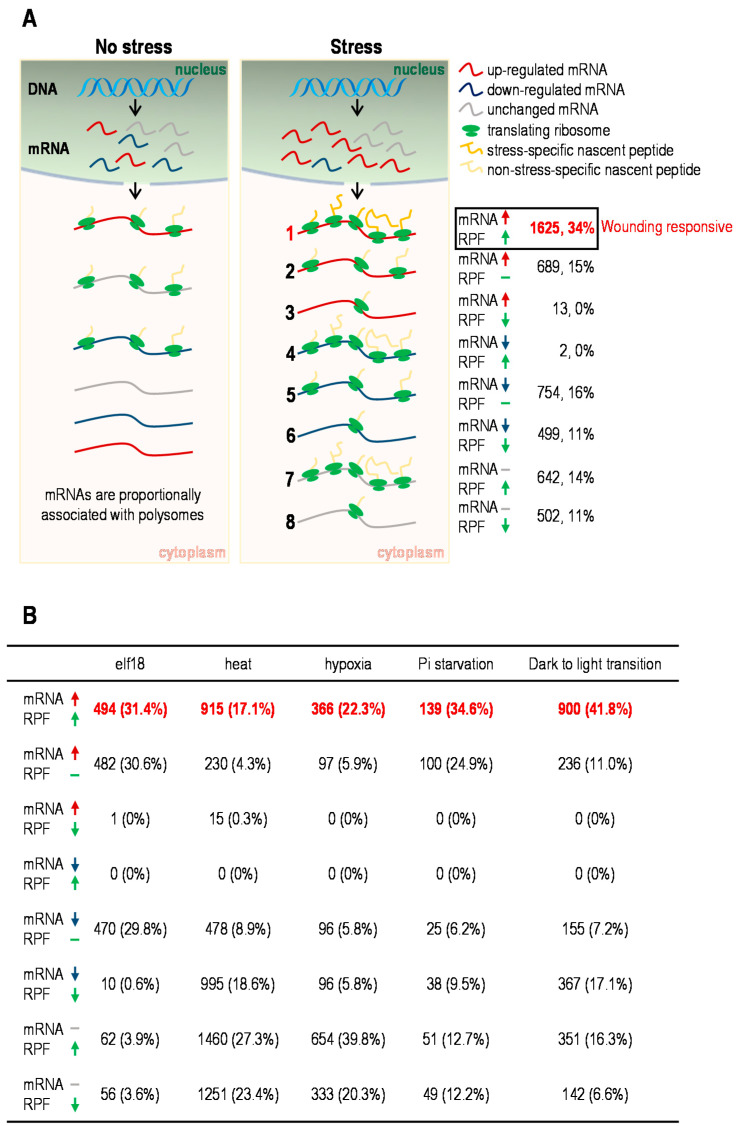
Schematic of stress-dependent translational control. (**A**) The genetic information stored in DNA leaves the nucleus as mRNA to be translated in the cytoplasm; red, blue and grey curves represent upregulated, downregulated and unchanged mRNA molecules respectively under specific stress such as wounding; 1~8 represent genes that undergo different types of transcriptional and/or translational regulation, total of which are the same set of genes as in Figure 1B excluding those with no significant changes at both levels. Gene numbers and proportions for each type of regulation under wounding are indicated on the right, among which there are 1625 (34%) typical wounding responsive genes with coordinately upregulated total mRNA and RPF termed type 1. (**B**) Gene numbers and proportions for each type of regulation under indicated stresses. Genes upregulated at both transcriptional and translational levels are critical for corresponding stress response (Figure 3C, Appendix A).

**Table 1 plants-10-00304-t001:** Data sets used in this study.

Stress	GEO Accession	References
elf18	GSE86581	[28]
heat	GSE69802	[25]
hypoxia	GSE50597	[26]
dark to light transition	GSE43703	[12]
Pi starvation	GSE98610	[27]
wounding	This study	Appendix A, Materials and Methods

## Data Availability

Raw Ribo-seq and RNA-seq data are available from NCBI with accession number PRJNA664417.

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
