# Peer review of "Preferential Ribosome Loading on the Stress-Upregulated mRNA Pool Shapes the Selective Translation under Stress Conditions"

_plants, 2021, doi:10.3390/plants10020304_

Round 1
Reviewer 1 Report
Authors properly revised their manuscript according to reviewers' comments.
I recommend the manuscript for publication as it is.
Reviewer 2 Report
Preferential ribosome loading on the stress-responsive mRNA pool shapes the translation profile under different stress conditions
In the above manuscript Chen et al. used publicly available data sets and self-generated data upon wounding to evaluate the relationships between the changes in transcriptome and the translatome in Arabidopsis thaliana upon stress exposure. Authors have performed several important and relevant analyses and presented their findings in an attractive manner. However, there are serious flaws with writing as pointed out in the numbered comments.
The manuscript was apparently gone through a previous round of review. Authors have taken care of the concerns of the reviewers to a certain extent, deposited the raw reads, and added a supplementary methods section including the codes used for the analyses. However, the original issues still exist with the use of inappropriate methods for certain analyses and methods not properly described in “Supplementary methods section” as described in the numbered comments below.
- A major flaw of this manuscript is its shallow writing style and interpretations based solely on the data presented in the manuscript itself. Most of the cited literature is in the introduction section and discussion section has only one reference. Many important related reviews and manuscripts were not cited and could be utilised to improve the scientific soundness of the manuscript. Some examples of such literature: PMID: 31004488, PMID: 25380596, PMID: 24375939, PMID: 27923997, PMID: 30920299, PMID: 19419242, PMID: 33294289. The current discussion section is a self-reflection of the results section. Therefore, the manuscript needs to be revised to include a discussion section with sound interpretations based on related literature.
- The main objective of this manuscript is to assess whether there is a common trend in the transcriptome-translatome relationship in different stresses rather than looking at shared genes among the different stresses. The authors started the manuscript with this objective in mind and performed some important analyses to answer this question. However, the major focus of the manuscript is transcriptome-translatome relationship during wounding stress and how it differs in other stresses. If the authors want to be true to their title and want to answer their original question, then they should slightly shift their focus and highlight trends during other stresses rather than confining them to supplementary figures. Moreover, discuss possible causes (technical or biological) for observed the differences in the transcriptome-translatome relationship under different stresses (also see comment 3 below for alternative analyses which might shed new light).
- The Venn-diagram in Figure 2A shows that 87% and 72% of the upregulated transcripts in ‘poly’ and ‘RPFs’ datasets respectively were also differentially upregulated in total mRNA data. It’s further evident in Figures S5 and S8. This suggests a significant positive correlation among the datasets for this subset of genes. This observation implies that performing correlation analyses between these total mRNA-polysome occupancy-ribosome loading (RPFs) data sets for up- and down-regulated genes separately may minimise method specific artefacts and may reveal meaningful correlations. Therefore, I would like to propose the authors for perform such an analysis.
- Pearson correlation analysis between samples
(a) According to “Supplemental Methods section 9” they were performed using ‘RPKM’ values (I guess Figures S2B-D since Figure 1A caption says that the authors used TPM). This is incorrect as pointed out in the first review. The reviewer#2 pointed out clearly that ‘RPKM’ should not be used to compare samples. Please use ‘cpm’ or ‘tpm’ for that. If you are not sure please read https://rna-seqblog.com/rpkm-fpkm-and-tpm-clearly-explained/ for an explanation why?
(b) In Figure 1A: Pearson correlation calculated with TPM but has only one sample each for ‘mRNA’, ‘poly’ and ‘RPF’ per group (control or wounded). Is TPM calculated per sample as in Supplementary methods section 8? What did the authors used for the correlation? This clearly demands that authors need to write a proper methods section detailing out how they performed each analysis to minimise any confusion.
(c) These high correlation values become even more questionable based on Figure S3F. Figure S3F is plotted with the RPFs-data. These read density plots around transcription ‘start’ and ‘end’ shows that “C1” (control 1) stand out from the rest of the samples. Please explain this disparity.
- Throughout the manuscript it is very unclear how the authors interpret the ‘polysome associated RNA-seq’ data (polysome occupancy of RNA). The mRNA should be loaded to ribosomes in order to end up in the polysomes. However, all ribosome-bound RNA may not end up in polysomes. Please be cautious of your interpretations. Following are two clear examples of that.
Example 1: In lines 100-107 the authors tried to propose the following as plausible explanation for the correlation differences between ‘mRNA’ vs. ‘poly’ (r=0.82) and ‘mRNA’ vs. ‘RPFs’ (r=0.62).
“…….indicating that upon wounding, transcribed mRNAs were proportionally engaged with the translation machinery, while ribosomes would not evenly load onto the polysome-associated mRNAs by the same proportion.”
If the polysome bound mRNA (‘poly’) data correlates better than RBFs data with total mRNA-data, then authors explanation of uneven polysome loading contradicts their own analysis.
Example 2: In lines 204-205 authors propose the following interpretation of their data:
“More importantly, reprogramming of gene expression is mainly orchestrated at
translational level by modulating ribosome loading but not polysome association.”
It is not clear what this statement biologically means.
- Authors have now provided a “Supplemental Methods” section mostly with snippets of R-code that they used for processing/analysing data. However, this is still not detail enough as pointed out below.
(a) They need to accompany a proper description of what the codes were used for or what were the associated figures since there is a lot of confusion throughout the manuscript as pointed out in comments 4a and 4b above and some examples below.
(b) How “Translational Efficiency (TE)” values were calculated were detailed in Figure 1 caption and are missing under Methods or Supplemental methods. Supplemental methods has section 11 with few lines of unannotated code without much information on TE calculation. In addition to the figure caption, Methods should include these details.
(c) Methods – Plant Growth and Treatments: How many plants per sample were used and how many leaves were wounded per plant? If the complete rosette is not wounded, then which leaf / leaves?
(d) The codes need to be properly annotated since they should be useful for ‘anyone’ who would like to repeat the analysis exactly similar to the authors’. Authors may annotate each line of code with “#”
E.g. “8. Calculation of TPM”
kb <- Count$Length/ 1000 #Length = Length of the genes
countdata <- Count[,3:8] # columns 3:8 have the count data. What are columns 1:3?
- GO-Term enrichment analysis qvalue cut-off of 0.1 is bit higher than usual (Supplemental Methods section 12). The general consensus cut-off is 0.05 or low. This value is not very high for me, but I would like the authors to highlight this fact in the main text and in the relevant figure captions. On a related note, functional enrichment analysis is more meaningful when it is performed separately for positively and negatively differentially expressed (up and down-regulated) genes as in Figure S8 rather than as in Figure 3B.
- In the abstract (lines 24-26) authors state “Moreover, Gene Ontology (GO) enrichment analysis showed that typical stress responsive genes were up-regulated at both transcriptional and translational levels, while stress irrelevant genes were changed solely at either level. What did the authors meant by “while stress irrelevant genes were changed solely at either level’ is not obviously clear? Reading the exactly similar comment in lines 221-213 in the manuscript suggests that the authors need to re write the statement to make the meaning explicitly clear.
Reviewer 3 Report
Manuscript entitled “Preferential ribosome loading on the stress-responsive mRNA pool shapes the translation profile under different stress conditions” by Chen et al describes a study of the response of Arabidopsis to several stresses using a comparative approach of the transcriptome and translatomes of stressed and control plants. The authors start this study using their self-generated omics data (plants exposed to wounding stress) and extend this comparative studies using publicly available omic datasets from other stresses (transition from dark to light, heat stress, hypoxia , Pi starvation as well as the response to the a pathogen-associated molecular pattern (elf18) exposure). The results obtained are very interesting, and they can add new data about the regulatory mechanisms implicated in the stress response in plants. The experimental design and the proposed approaches used in this work seem both correct for the most part. However, in my opinion, some aspects of how the data obtained is discussed need to be revised throughout the manuscript before considering this work suitable for publication:
As a general comment, the discussion of work in this version of the manuscript is often rather superficial and should be improved. The authors are too inclined to repeat the description of the results again, instead of integrating them into previous results from the literature that helps to arrive at some new interpretation regarding the observed results. In this sense, I must make the authors two recommendations:
- To try to place their results in the context of a broader framework. They should make a more exhaustive search of previous works regarding studying other plant growth and development processes apart from stress where it has been described or explored the possible overlap of transcriptional and translational regulation suggested by this work.
- The sets of genes detected in the different comparisons are also discussed in a very superficial way. The authors should try to further discuss the biological implications of the proteins encoded for the different genes detected. For example, I believe that the authors should try to describe better what is the implication in the response to stress of the set "core stress responsive genes" detected in the different comparisons described in table S3.
Other comments about your manuscript:
Line 230: please correct “(A)Tanslational efficiency” with ““(A)Translational efficiency”
Line 235-236 and figure S7, S8 legends: please consider changing “Functional themes” with “Functional categories”
Line 222-225: Stress is a specific situation, not a process that “prefers” to do a particular task or process. Please consider changing the sentence to a more accurate expression. Perhaps something like” This makes explicit the concept that the increment of both mRNA and RPF levels for a specific set of genes seems to be a common mechanism to ensure a more targeted response against each stress”
Line 274: please correct “tanslatomes” with “translatomes”
Line 302: 150 microliters or micrograms of pulverized frozen tissue?
Supplementary material, figures S2B-D: In addition to the Pearson correlation representations, included I think that perhaps a Principal Component Analysis (PCA) graph should be included to check if the replicates of the transcriptomes/translatomes from each condition cluster together and in that way help to exclude the possibility of outlier replicas that could tamper your posterior comparative analyses.
Round 2
Reviewer 2 Report
Authors have addressed the concerns and/or provided satisfactory explanations.
Reviewer 3 Report
I am happy to see that the revised version of the manuscript presents a considerable improvement, and that the main issues pointed previously have been addressed. Still, I feel that several minor issues should be addressed:
Line 29: I think the term "detoxify" is not a term that can be applied in a general way to refer to the plant response to wounding stress. I suggest to the authors to modify the sentence to something like “typical stress responsive genes are generally involved in functional categories related to deal with the deleterious effects caused by the imposed wounding stress, such as response to wounding, response to water deprivation and response to jasmonic acid”
Line 115: RPF abbreviation has already been defined in line 62
Line 309: genes are segments of DNA. By themselves, they do not act directly on any process: they code for proteins that participate in the processes carried out in cells. For this reason, I recommend that the authors modify the phrase "appreciated genes that may act as key driving force of the multiple stress response" with a more appropriate expression.
Line 379: I think that a future reader might need a definition of "hub genes" in this sentence.
Supplementary figure S2 legend: PCA and TPM abbreviations should be defined in the legend.
Author Response
Please see the attachment.

This manuscript is a resubmission of an earlier submission. The following is a list of the peer review reports and author responses from that submission.
Round 1
Reviewer 1 Report
In this study, Chen et al. compared the transcriptome and translatome using public datasets and their own data. They found that changes in transcriptome and translatome are highly correlated in dark to light transition or heat stress while they did not find such correlation in the four stress conditions. In addition, stress responsive genes were up-regulated at both transcriptional and translational levels, while stress irrelevant genes were changed solely at either level. Overall, the manuscript is very nicely written. The quality of the manuscript including data analyses, figures, and tables is excellent. However, the length of the manuscript is relatively short.
Comment 1. I recommend including more background associated with this study in introduction.
Comment 2. In discussion, please compare other similar studies that conducted comparison between transcriptome and translatome.
Comment 3. Moreover, I did not find accession numbers of RNA-seq from this study (wounding). Please deposit NGS data in public database such as SRA and provide accession numbers in the manuscript.
Author Response
Thank you very much for your hard work in processing our manuscript. We have carefully read the comments and revised the manuscript as suggested. Below are our point by point response. Also please see the attachment.
1. Comment 1. I recommend including more background associated with this study in introduction.
We have revised the introduction to include more details (lines 44-48, lines56-60).
2. Comment 2. In discussion, please compare other similar studies that conducted comparison between transcriptome and translatome.
Another study conducting comparison between transcriptome and translatome have been included in the discussion, which was published online recently in the journal Nature (lines 272-276).
3. Comment 3. Moreover, I did not find accession numbers of RNA-seq from this study (wounding). Please deposit NGS data in public database such as SRA and provide accession numbers in the manuscript.
Accession number can now be found in the section of Data availability (lines 347-350). We have released the data and maybe it will be available several days later.

Reviewer 2 Report
The authors present a study about the connection of mRNA, Polysomes and translational activity under different stress conditions to get a deeper insight into regulatory measures in Arabidopsis plants in answer to several stresses. This is a highly important topic across all organisms, since much too often RNA-Seq data are used to define and interpret regulatory pathways when RNA does not really represent what is going on at protein level.
To this end, they performed wounding experiments themselves, whereas for the other stresses made use of publicly available data. I have to say that I am not well versed in statistical analyses, but their data and conclusions were easy to follow and they did convince me. The step taking this kind of research to a completely different level would of course include quantitative proteomics, however I am completely aware of the complexity and cost of such an endeavor. Therefore, I really can’t find any fault with this manuscript.
Author Response
Thank you very much for your hard work in processing our manuscript. Please see the attachment.
